# Outcomes of in-hospital cardiac arrest managed with and without a specialized code team: A retrospective observational study

Yasmeen Abu Fraiha[1]*, Tali Shafat[2], Shlomi Codish[3], Amit Frenkel[4], Dror Dolfin[3], Jacob Dreiher[3], Yuval Konstantino[5], Said Abu Abed[6], Doron Schwartz[3], Alexander Fichman[4], Luba Kvich[3], Ori Galante[7]

1 Department of Internal Medicine B, Soroka University Medical Center and Faculty of Health Sciences, Ben-Gurion University of the Negev, Be'er Sheva, Israel, 2 Infectious Diseases Unit, Soroka University Medical Center and Faculty of Health Sciences, Ben-Gurion University of the Negev, Be'er Sheva, Israel, 3 Medical Administration, Soroka University Medical Center and Faculty of Health Sciences, Ben-Gurion University of the Negev, Be'er Sheva, Israel, 4 Surgical Intensive Care Unit, Soroka University Medical Center and Faculty of Health Sciences, Ben-Gurion University of the Negev, Be'er Sheva, Israel, 5 Department of Cardiology, Soroka University Medical Center and Faculty of Health Sciences, Ben-Gurion University of the Negev, Be'er Sheva, Israel, 6 Pediatrics Department, Soroka University Medical Center and Faculty of Health Sciences, Ben-Gurion University of the Negev, Be'er Sheva, Israel, 7 Medical Intensive Care Unit, Soroka University Medical Center and Faculty of Health Sciences, Ben-Gurion University of the Negev, Be'er Sheva, Israel

* Yasmeenaf23@gmail.com

**Data Availability Statement:** Data cannot be shared publicly because it is owned by Soroka hospital and Clalit Health Services. If needed after

## Abstract

### Background

In-hospital cardiac arrest (IHCA) still has a poor prognosis despite medical advancements in recent decades. Early and high-quality cardiopulmonary resuscitation (CPR), as well as good teamwork, are important prognostic factors. There are no clear guidelines regarding the composition of a dedicated hospital CPR team. We compared outcomes of IHCA treated by a dedicated hospital CPR team compared to ward medical staff with advanced cardiac life support (ACLS) training.

### Methods

A single-center retrospective observational study based on the cardiopulmonary resuscitation database of Soroka University Medical Center from January 2016 until December 2019. We compared the results of resuscitations conducted by regular ward medical staff, certified in ACLS, versus those conducted by the dedicated hospital's CPR team.

### Results

Of the 360 CPR events analyzed, 141 (39.1%) ended in return of spontaneous circulation, 70 (19.4%) patients were alive after 24 hours, 23 (6.4%) survived for 30 days, and 18 (5%) survived to discharge. Of those who survived to discharge, 11 (61.1%) had a cerebral performance category (CPC) score of 1–2, and 7 (38.9%) had a score of 3–4 (mean 2.09). Survival-to-discharge was significantly higher in the CPR-team group compared to the ward-

publication, data can be restrictly available only after approval of the Institutional Ethics Committee (contact via corresponding author) for researchers who meet the criteria for access to confidential data. For further information. please contact Ms. Carolina Havivian, Ethics Committee Liaison, CarolinaHv@clalit.org.il

**Funding:** The author(s) received no specific funding for this work.

**Competing interests:** The authors have declared that no competing interests exist.

team group (7.6% vs. 1.9%, p = 0.013). However, with propensity score analysis the difference in survival became insignificant (RR = 1.97, 95% CI: 0.40–9.63, p = 0.40).

## Conclusion

We found no difference in survival between IHCA treated by a dedicated hospital CPR team compared to a standard ward team, both trained with biennial ACLS training. Nevertheless, crude survival-to-discharge was significantly higher in the CPR-team group.

## Introduction

In-hospital cardiac arrest (IHCA) occurs in over 290,000 adults each year in the United States [1]. Despite improvements in implementing high-quality cardiopulmonary resuscitation (CPR), survival-to-discharge rates from IHCA continue to be 15–30% [1–5]. Although mortality is strongly related to underlying medical conditions, inappropriate resuscitation and human errors are likely to contribute to decreased survival. Failure to adhere to Adult Cardiac Life Support (ACLS) protocol is associated with decreased rates of return of spontaneous circulation (ROSC) and of survival to hospital discharge [6–10].

In 2013, the American Heart Association consensus statement recommended that providers utilize a coordinated team response with "specific role responsibilities" [11]. However, the guidelines did not state the specific roles and duties of the team, nor the team members' qualifications and the required training. Furthermore, according to the latest focused update of the guidelines (2019), there is insufficient evidence to make a specific recommendation about the ideal frequency of the providers' retraining [12].

In recent years, a small number of institutions published data on their IHCA results, focusing on team interventions rather than protocol interventions. Qureshi et al. showed that a standardized CPR team, comprised of an emergency physician and critical care nurses, that responded to all CPR events, eliminated the variation in achieving ROSC between daytime and nighttime, as well as between weekdays and weekends [13]. However, they did not report data regarding 24-h survival, survival-to-discharge, and neurological outcomes. Spitzer et al. implemented an in-hospital pit-crew model with an emphasis on leadership and communication skills but failed to show significant results due to a small sample size [6]. Sodhi et al. [14] reformulated all of their resuscitations protocols and reframed the CPR team to always include an advanced cardiovascular life support (ACLS) provider. These interventions led to a significant increase in survival-to-discharge rates, but this increase could not be associated specifically with team adjustments. Hence, the effect of a standardized, specified, and well-trained in-hospital CPR team still needs to be determined. This study aims to explore the difference in outcomes of IHCA patients treated by general wards' medical staff compared to the dedicated hospital CPR team.

## Materials and methods

### Study population and design

In this single-center retrospective observational study, we included all consecutive hospitalized patients who underwent CPR due to IHCA at Soroka University Medical Center (SUMC) internal medicine wards during the period between January 1st, 2016 and December 31st, 2019, and were registered in the institutional electronic database. The study protocol was approved

by the Institutional Ethics Committee (Soroka Medical Center Ethics committee approval number 0074-20-SOR).

IHCA, as opposed to death without resuscitation, is defined in this study as the loss of circulation prompting resuscitation with chest compressions, defibrillation, or both. All in-hospital CPRs are reported and recorded in hospital records, which include the documentation of the treating team, its arrival time, CPR time of day, drugs administered during CPR, procedures performed during resuscitation, and outcomes. The CPR team at SUMC is comprised of a cardiologist (either a fellow or an attending), an anesthesiologist (either a resident or an attending), and a qualified nurse, all of whom are ACLS certified. The cardiologist is the dedicated team leader, given their expertise in cardiac arrest management; the anesthesiologist is the airway manager, given their expertise in intubation and airway management; and the qualified nurse is responsible for drug administration and recording.

In Israel, as of 2006 (and revised again in March 2019), the Israeli Ministry of Health directs hospitals to assign designated institutional CPR teams [15]. All team members must be certified as ACLS providers. The ACLS certification process includes passing one full course that complies with the formal requirements and guidelines of the American Heart Association (AHA) and the International Liaison Committee on Resuscitation (ILCOR), lasting at least 14 hours, in which at least 50% of the time is dedicated to practice simulations. All ACLS certified providers are required to undergo biennial retraining course, lasting no less than 8 hours. Failing to undergo a biennial retraining course requires passing a full 14-hour course once again.

All Israeli physicians are required to pass an ACLS full course before the beginning of their clinical work as interns. However, most of them will not continue with the retraining. The Israeli Ministry of Health requires biennial retraining only from physicians who work in settings with high-risk patients, including intensive care, emergency medicine, cardiology, internal medicine, anesthesiology, dialysis and operating rooms). However, there is no direct obligation to activate the dedicated CPR team during IHCA. As a result, CPR managed by internal medicine ward teams is also managed by certified ACLS providers, but with different expertise, experience (interns and residents) and no formal role assignments, as opposed to the dedicated CPR team.

### Inclusion and exclusion criteria

Due to the fact that all internal medicine physicians are certified ACLS providers, and are required to complete biennial ACLS retraining, they may choose whether or not to call the hospital's CPR team. Therefore, we included in the analysis all the consecutive records of adult patients (≥18 years old) who underwent CPR due to IHCA in the internal medicine wards. We excluded all patients who suffered from out-of-hospital cardiac arrest, CPRs that were managed in wards other than the internal medicine wards (in which physicians are mostly BLS providers and are obligated to call the hospital's CPR team), and CPRs that were managed by a single CPR team member (e.g., only a cardiologist / anesthesiologist) combined with the regular ward medical team.

### Captured data

Clinical and laboratory data were retrieved from patients' electronic medical records. Parameters collected included demographic information, admission date, discharge date, admission unit, transfers during hospitalization, primary admission diagnosis, CPR date, lab results during the hospitalization, CPR parameters and procedures, medical staff participating in CPR, and post-CPR information including duration of CPR, ROSC (yes/no), 24-h survival (yes/no), 30-days survival (yes/no), survival-to-discharge (yes/no), and Pittsburgh Cerebral

Performance Category (CPC) score at discharge. Hospital discharge was defined as the cessation of hospitalization by either discharge home or to a rehabilitation facility.

Due to the retrospective nature of this study, and in order to deal with the possibility of selection bias when deciding whether or not to call the hospital's dedicated CPR team based on comorbidities, we assessed the patients' physical status, comorbidities and functional abilities by calculating Charlson Comorbidity Index [16], Norton Score [17], and Morse Fall Scale [18] per patient on admission day and on the day of IHCA. Charlson Comorbidity Index and Norton Score were shown to have a prognostic value on mortality in the hospital setting [19, 20], while Morse Scale is associated with worse prognosis in specific situations such as cardiac rehabilitation [21] and long-term care [22].

## Medical scales and indices

In order to take patients' comorbidities into account in the outcome analysis, we have calculated the following scores for each patient: Charlson Comorbidity Index [16], Norton Score [17], and Morse Fall Scale [18].

The Charlson Comorbidity Index is a well-established prediction tool that predicts ten-year survival probability based on past and current medical conditions including age and major comorbidities. The maximum possible score is 37, although any score above 6 points predicts 0% ten-year survival probability. The acceptable cut-off index in the known literature is 4 points, as it indicates 53% ten-year survival.

The Norton Scoring System is used to assess the risk of developing pressure ulcers and takes into account the patient's general physical condition, mental condition, ambulatory activity, mobility, and incontinence. The scores range from 5 to 20, while 5 points indicate maximal risk and 20 points indicate minimal risk.

The Morse Fall Scale is a simple method to assess the patient's risk for falling during hospitalization. It consists of a history of falling (immediate or within 3 months), any secondary diagnosis, need for ambulatory aid, administration of intravenous medications, quality of gait, and mental status. The scale range is 0–125, with 0–24 points indicating no risk, 25–50 points indicating low risk, and any score above 50 indicating a high risk.

## Outcomes

The primary outcome was survival-to-discharge. Cerebral functional status was determined at hospital discharge according to the Pittsburgh Cerebral Performance Category (CPC). CPC score of 1–2 was considered good neurologic outcome and poor neurologic outcome was defined as CPC 3–5 [23]. Secondary outcomes were ROSC, 24-hour survival, 30-day survival, and survival-to-discharge.

## Statistical analysis

The statistics for continuous variables included mean, standard deviation, minimum, maximum, and sample. Categorical variables were described with numbers and percentages. The t-test was used for comparison of continuous variables and chi-square or Fisher's exact tests were used for categorical data. We used the Mann-Whitney test for the comparison of variables with non-normal distribution.

We used propensity score (PS) weighting technique to balance the distribution of covariates between patients who were treated by CPR team vs ward team. This technique assigns weights to each observation based on their propensity scores, which are the predicted probabilities of being assigned to the treatment group (CPR team call) based on their covariates. The weights

are then used in the analysis to account for differences in covariate distributions between treatment groups.

The PS, which was estimated using a non-parsimonious logistic regression model, was defined as the probability of the patients being included in the CPR team group based on the observed baseline characteristics. Modeling variables in the logistic regression analysis included age, gender, ethnicity, background diagnoses of stroke, dementia or malignancy, Charlson Comorbidity Index (CCI) ≥4, admission Norton score ≥15, CPR-day Norton score≥15, mechanical ventilation before CPR, vasopressor treatment before CPR, initial physician seniority (attending/resident/intern), CPR rhythm (shockable/non-shockable), and the time-of-day that the CPR was taking place (morning/evening/night).

Poisson regression was utilized to calculate rate ratio (RR) of PS-weighted CPR team on survival-to-discharge. A two-tailed $P$-value of ≤0.05 was considered significant. All p-values reported are rounded to three decimal places. All statistical analyses were conducted using SPSS 25.0 statistical software (IBM Corp Armonk, NY, USA) and R 4.2, R studio R Core Team (2022). Vienna, Austria.

## Results

Of 452 records of CPRs documented between January 2016 to December 2019, 360 records were included for analysis (Fig 1). The mean age was 73.2 and 57% were male. Eighteen patients underwent two separate CPRs, only their first recorded CPR was included for analysis. Of the 360 CPRs, 141 (39.2%) ended in ROSC, 70 (19.4%) patients were alive after 24 hours, 23 (6.4%) survived for 30 days and 18 (5%) survived to discharge. Of those who survived to discharge, 11 (61.1%) had a CPC score of 1–2 and 7 (38.9%) had a score of 3–4. 198 CPRs (55%) were managed by the on-call CPR team and 162 (45%) were managed by the ward's team.

Both groups had a similar prevalence of comorbid conditions, including ischemic and congestive heart diseases, hypertension, diabetes, malignancies, chronic kidney disease, dementia, and major surgeries (Table 1). A notable exception is the Norton score, which was significantly lower in patients who had their CPR managed by the ward team (14 vs. 16, p<0.001). Table 2 shows baseline characteristics of patients who survived to discharge compared to patients who did not survive. There were significantly higher percentage of male gender in the survivors' group (72% vs. 55.5%, p = 0.035). The mean age of survivors was significantly lower than that of non-survivors (64.9 vs. 74.0, p<0.0001). In addition, the survivors' group has a significantly lower Charlson score (6 vs. 7, p = 0.027) and higher Norton score (17 vs. 15, p = 0.008).

Table 3 describes patients' parameters during CPR. 79.7% of CPRs were initiated by department residents or interns, who called the CPR team in 61.7% of cases, while attending physicians called the CPR team in only 28.8% of cases.

Generally, the reported IHCAs were evenly distributed throughout the day. However, there was a significant difference in the distribution among the groups. Most CPRs managed by ward teams were performed during morning shifts (7am-3pm, 44.4%), while most CPRs managed by the institutional CPR team were performed during night shifts (11pm-7am, 46.5%), (p<0.001). Significantly more patients treated by the ward medical staff were already on mechanical ventilation at the time of IHCA (20.4% vs. 5.1%, p<0.001).

Table 4 shows crude survival outcomes. 96 patients achieved ROSC in the CPR team group (48.5%) compared to the ward team group (27.5%, p<0.0001; OR 2.45, 95% CI 1.57–3.81). Survival-to-discharge rate was also significantly higher in patients treated by the hospital's CPR team (7.6% vs. 1.9%, p<0.013; OR 4.34, 95% CI 1.24–15.28). However, there was no significant difference in survival rates after 24 hours (22.7% vs. 15.4%, p = 0.082; OR 1.61, 95% CI

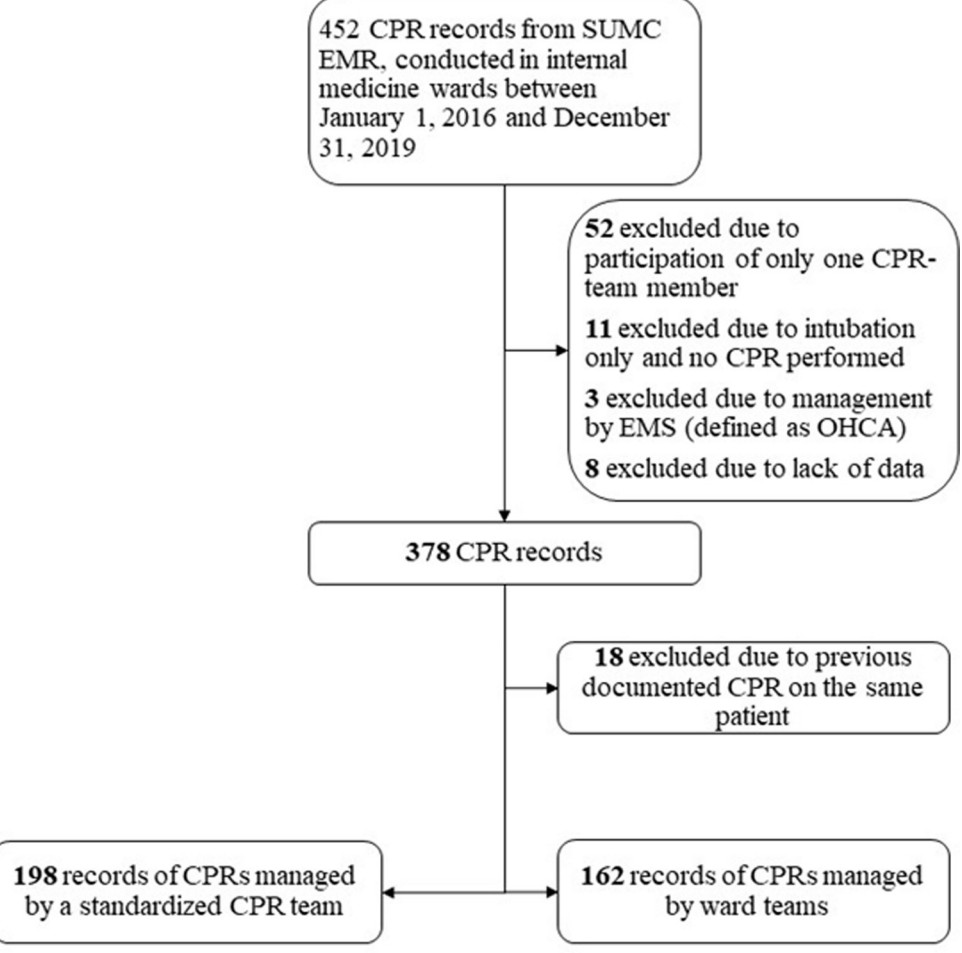

**Fig 1. Population selection.** The figure describes the patients' selection process. CPR = cardiopulmonary resuscitation; SUMC = Soroka University Medical Center; EMR = electronic medical records; EMS = emergency medical services; OHCA = out-of-hospital cardiac arrest.

0.94–2.77) and after 30 days (8.6% vs. 3.7%, p = 0.06; OR 2.44, 95% CI 0.94–6.35), nor in length of stay (median of six days for both groups, p = 0.556) and CPC score (4% vs. 1.9%, p = 0.23; OR 2.23, 95% CI 0.58–8.55), although it seems that the general trend is towards better outcomes in the CPR team group.

The propensity score for the probability of calling the CPR team per patient characteristics was built using logistic regression (Table 5 and Figs 2 and 3). RR of survival-to-discharge according to PS-weighted CPR team call was not statistically significant (RR = 1.97, 95% CI: 0.40–9.63, p = 0.40).

## Discussion

The main finding of this study is that an institutional designated CPR team did not have better CPR outcomes compared to ward teams who are trained ACLS providers. This may imply that with strict ACLS certification process and retraining, a dedicated in-hospital CPR team adds little to no improvement in IHCA outcomes. This is in concordance with similar studies performed on out-of-hospital cardiac arrest [24, 25] and the pediatric population [26], and with increasing evidence supporting team retraining [27, 28], minimizing interruptions in chest

**Table 1. Baseline characteristics of the study population–CPR team vs. ward team (N = 360).**

| Variable N (%) | | CPR Team (N = 198) | Ward Team (N = 162) | P Value |
|---|---|---|---|---|
| Age, years (mean± SD) | | 73 ± 13.1 | 75 ±12.8 | 0.148 |
| Age > 75 years | | 104 (52.5) | 95 (58.6) | 0.246 |
| Gender | Male | 119 (60.1) | 87 (53.7) | 0.222 |
| Ethnicity | Jewish | 172 (87.3) | 140 (86.4) | 0.804 |
| | Bedouin[*] | 25 (12.7) | 22 (13.6) | |
| Chronic ischemic heart disease | | 91 (46.0) | 65 (40.1) | 0.266 |
| Congestive heart failure | | 102 (51.5) | 75 (46.3) | 0.324 |
| Essential hypertension | | 132 (66.7) | 111 (68.5) | 0.709 |
| Diabetes mellitus | | 119 (60.1) | 94 (58.0) | 0.690 |
| Dyslipidemia | | 120 (60.6) | 96 (59.3) | 0.795 |
| Valvular disease | | 103 (52.0) | 74 (45.7) | 0.231 |
| Obesity | | 76 (38.4) | 55 (34.0) | 0.384 |
| Arrhythmia | | 64 (32.3) | 42 (25.9) | 0.185 |
| Malignancy | | 43 (21.7) | 38 (23.5) | 0.694 |
| Chronic kidney disease | | 58 (29.3) | 53 (32.7) | 0.484 |
| Hemodialysis | | 15 (7.6) | 12 (7.4) | 0.952 |
| Peripheral vascular disease | | 62 (31.3) | 48 (29.6) | 0.730 |
| Chronic lung disease | | 52 (26.3) | 31 (19.1) | 0.110 |
| Cerebrovascular accident | | 35 (17.7) | 42 (25.9) | 0.058 |
| Peptic ulcer disease | | 37 (18.7) | 30 (18.5) | 0.967 |
| Dementia | | 25 (12.6) | 30 (18.5) | 0.122 |
| Connective tissue disease | | 23 (11.6) | 24 (14.8) | 0.370 |
| Liver disease | | 23 (11.6) | 14 (8.6) | 0.355 |
| Pacemaker | | 18 (9.1) | 15 (9.3) | 0.956 |
| Surgery | | 83 (41.9) | 77 (47.5) | 0.286 |
| Charlson index (median, inter-quartile range) | | 7, 5–8 | 7, 5–9 | 0.227 |
| Charlson index ≥4 | | 180 (90.9) | 151 (93.2) | 0.425 |
| Norton (admission) (median, inter-quartile range) | | 16, 13–18 | 14, 11–17 | 0.001 |
| Norton (admission) ≥15 | | 114 (58.8) | 75 (47.8) | 0.040 |
| Morse fall scale (admission) (median, inter-quartile range) | | 50, 40–70 | 55, 40–75 | 0.245 |
| Morse fall scale (admission) ≥25 | | 176 (90.7) | 141 (89.8) | 0.774 |
| ICU admission any time during hospitalization (prior to CPR) | | 38 (19.2) | 21 (13.0) | 0.112 |
| Vital signs and lab results on admission | Tachycardia (>100 bpm) | 67 (34.0) | 53 (32.9) | 0.828 |
| | SBP < 90 mmHg | 12 (6.1) | 16 (9.9) | 0.178 |
| | $O_2$ saturation <90% | 42 (21.3) | 33 (20.6) | 0.873 |
| | Fever (>38.0˚C) | 24 (12.4) | 29 (18.4) | 0.123 |
| | Anemia (<12 g/dL) | 150 (76.1) | 122 (75.3) | 0.854 |
| | Leukocytosis (>11 k/mm$^3$) | 90 (45.7) | 79 (48.8) | 0.561 |
| | Leukopenia (<4.5 k/mm$^3$) | 8 (4.1) | 11 (6.8) | 0.250 |
| | Thrombocytopenia (<150 k/ul) | 28 (14.2) | 34 (21.0) | 0.091 |
| | Creatinine >1.2 mg/dL | 119 (60.4) | 90 (55.6) | 0.354 |
| | Urea > 43 mg/dL | 152 (77.2) | 133 (82.1) | 0.249 |
| | ALP > 120 IU/L | 82 (43.9) | 71 (45.8) | 0.717 |
| | AST > 33 IU/L | 70 (43.2) | 52 (40.3) | 0.618 |
| | ALT > 35 IU/L | 44 (23.5) | 45 (29.0) | 0.248 |
| | GGT > 39 IU/L | 100 (53.5) | 94 (60.6) | 0.183 |
| | pH<7.35 (N = 278) | 91 (59.5) | 81 (64.8) | 0.363 |

*(Continued)*

**Table 1.** (Continued)

| Variable N (%) | | CPR Team (N = 198) | Ward Team (N = 162) | P Value |
|---|---|---|---|---|
| | $PCO_2 > 50$ mmHg (N = 278) | 62 (40.8) | 62 (49.6) | 0.142 |
| | $HCO_3^- < = 20$ (mEq/L) | 29 (19.0) | 34 (27.2) | 0.102 |

This table demonstrates the absolute numbers of patients with each characteristic collected from the electronic medical records. The number in brackets represents the percentage of patients out of N, N being the total number of patients in the group (CPR team vs. ward team). P-value was calculated for each variable when comparing the groups and is shown here. CPR = cardiopulmonary resuscitation; SD = standard deviation; ICU = intensive care unit; SBP = systolic blood pressure; BUN = blood urea nitrogen.

[*] The two largest ethnic populations treated in Soroka University Medical Center are Jews and Bedouins (Arabs).

**Table 2. Baseline characteristics of the study population–Survivors vs. non-survivors (N = 360).**

| Variable N (%) | | Survivors (N = 18) | Non-survivors (N = 342) | P Value |
|---|---|---|---|---|
| Age, years (mean± SD) | | 65.8 ± 14.3 | 74.6 ± 12.7 | **0.005** |
| Gender | Male | 10 (55.6) | 196 (57.3) | 0.883 |
| Ethnicity | Jewish | 16 (88.9) | 296 (86.8) | 0.798 |
| | Bedouin | 2 (11.1) | 45 (13.2) | |
| Chronic ischemic heart disease | | 6 (33.3) | 150 (43.9) | 0.380 |
| Congestive heart failure | | 7 (38.9) | 170 (49.7) | 0.371 |
| Essential hypertension | | 9 (50.0) | 234 (68.4) | 0.104 |
| Diabetes mellitus | | 11 (61.1) | 202 (59.1) | 0.863 |
| Dyslipidemia | | 10 (55.6) | 206 (60.2) | 0.693 |
| Valvular disease | | 9 (50.0) | 168 (49.1) | 0.942 |
| Obesity | | 5 (27.8) | 126 (36.8) | 0.436 |
| Arrhythmia | | 6 (33.3) | 100 (29.2) | 0.710 |
| Malignancy | | 4 (22.2) | 77 (22.5) | 0.977 |
| Chronic kidney disease | | 5 (27.8) | 106 (31.0) | 0.773 |
| Hemodialysis | | 1 (5.6) | 26 (7.6) | 0.748 |
| Peripheral vascular disease | | 4 (22.2) | 106 (31.0) | 0.431 |
| Chronic lung disease | | 6 (33.3) | 77 (22.5) | 0.288 |
| Cerebrovascular accident | | 2 (11.1) | 75 (21.9) | 0.275 |
| Peptic ulcer disease | | 6 (33.3) | 61 (17.8) | 0.100 |
| Dementia | | 3 (16.7) | 52 (15.2) | 0.867 |
| Connective tissue disease | | 3 (16.7) | 44 (12.9) | 0.641 |
| Liver disease | | 2 (11.1) | 35 (10.2) | 0.905 |
| Pacemaker | | 1 (5.6) | 32 (9.4) | 0.586 |
| Surgery | | 9 (50.0) | 151 (44.2) | 0.626 |
| Charlson index (median, inter-quartile range) | | 5 (1–7) | 7 (5–9) | 0.015 |
| Charlson index ≥4 N (%) | | 12 (66.7) | 319 (93.3) | <0.001 |
| Norton (admission) (median, inter-quartile range) | | 18 (12–19) | 15 (12–17) | 0.026 |
| Norton (admission) ≥15 N (%) | | 12 (66.7) | 177 (53.2) | 0.263 |
| Morse fall scale (admission) (median, inter-quartile range) | | 45 (38–62) | 55 (40–72) | 0.329 |
| Morse fall scale (admission) ≥25 N (%) | | 16 (88.9) | 301 (90.4) | 0.834 |

This table describes the absolute numbers of patients with each characteristic collected from the electronic medical records. The number in brackets represents the percentage of patients out of N, N being the total number of patients in the group (survivors vs. non-survivors).

SD = standard deviation.

**Table 3. Parameters at time of CPR (N = 360).**

| Variable N (%) | | CPR Team (N = 198) | Ward Team (N = 162) | P Value |
|---|---|---|---|---|
| Physician initiating CPR | Intern | 10 (5.1) | 0 (0) | <0.001 |
| | Resident | 167 (84.3) | 110 (67.9) | |
| | Attending | 21 (10.6) | 52 (32.1) | |
| CPR during first 24 hours of admission | | 36 (18.2) | 25 (15.4) | 0.489 |
| CPR time of day | 23:00–07:00 | 92 (46.5) | 39 (24.1) | <0.001 |
| | 07:00–15:00 | 37 (18.7) | 72 (44.4) | |
| | 15:00–23:00 | 69 (34.8) | 51 (31.5) | |
| Time from admission to CPR, in days (median, inter-quartile range) | | 3, 1–9 | 5, 1–11 | 0.085 |
| Norton score (median, inter-quartile range) | | 15, 12–18 | 14, 10–16 | 0.001 |
| Norton ≥15 | | 111 (56.1) | 67 (41.6) | 0.006 |
| Morse fall scale (median, inter-quartile range) | | 50, 40–70 | 55, 40–75 | 0.203 |
| Morse fall scale ≥25 | | 179 (90.4) | 147 (91.3) | 0.769 |
| Mechanical ventilation | | 10 (5.1) | 22 (20.4) | <0.001 |
| Vasopressors | | 18 (9.1) | 22 (13.6) | 0.178 |
| CPR duration, in minutes (median, inter-quartile range) | | 30, 20–37 | 27, 20–37 | 0.446 |
| Initial rhythm during CPR | Asystole | 109 (55.1) | 99 (61.1) | 0.101 |
| | Pulseless electrical activity | 64 (32.3) | 36 (22.2) | |
| | Ventricular fibrillation / pulseless ventricular tachycardia | 11 (5.6) | 12 (7.4) | |
| | Other | 14 (7.1) | 15 (9.3) | |
| Last vital signs and lab results on CPR day (prior to CPR) | Tachycardia (>100 bpm) | 65 (32.8) | 46 (28.4) | 0.365 |
| | SBP < 90 mmHg | 32 (16.2) | 43 (26.5) | 0.016 |
| | $O_2$ saturation <90% | 43 (21.8) | 39 (24.1) | 0.614 |
| | Fever (>38.0°C) | 17 (8.6) | 27 (16.7) | 0.021 |
| | Anemia (<12 g/dL) | 157 (79.7) | 135 (83.3) | 0.379 |
| | Leukocytosis (>11 k/mm$^3$) | 98 (49.7) | 88 (54.7) | 0.355 |
| | Leukopenia (<4.5 k/mm$^3$) | 9 (4.6) | 8 (5.0) | 0.859 |
| | Thrombocytopenia (<150 k/ul) | 42 (21.3) | 40 (24.8) | 0.430 |
| | Creatinine >1.2 mg/dL | 122 (61.6) | 103 (63.6) | 0.702 |
| | Urea > 43 mg/dL | 161 (81.3) | 141 (87.0) | 0.142 |
| | ALP > 120 IU/L | 79 (42.0) | 73 (45.9) | 0.467 |
| | AST > 33 IU/L | 89 (50.6) | 72 (47.7) | 0.603 |
| | ALT > 35 IU/L | 63 (33.5) | 54 (34.0) | 0.929 |
| | GGT > 39 IU/L | 115 (61.2) | 111 (69.8) | 0.092 |
| | pH<7.35 (N = 278) | 102 (67.1) | 88 (65.7) | 0.798 |
| | $PCO_2$ > 50 mmHg (N = 278) | 73 (48.0) | 63 (47.0) | 0.864 |
| | $HCO_3^-$ < = 20 (mEq/L) | 53 (34.9) | 44 (33.1) | 0.751 |
| **Interventions during CPR** | **Chest compressions** | **197 (99.5)** | **160 (98.8)** | **0.449** |
| | **Ambu bag / non-invasive mechanical ventilation** | **194 (98.0)** | **154 (95.1)** | **0.125** |
| | **Defibrillation** | **37 (18.7)** | **29 (17.9)** | **0.848** |
| | **IV access** | **96 (48.5)** | **58 (35.8)** | **0.016** |
| | **Intubation** | **169 (89.4)** | **79 (61.2)** | **<0.001** |
| | **Pacing** | **3 (1.5)** | **0** | **0.116** |
| | **IO** | **0** | **1 (0.6)** | **0.268** |

This table describes the absolute numbers of patients with each characteristic collected from the electronic medical records. The number in brackets represents the percentage of patients out of N, N being the total number of patients in the group (CPR team vs. ward team). For continuous variables, a limit was set and the absolute number was counted accordingly. P-value was calculated for each variable when comparing the groups, and is shown here.

CPR = cardiopulmonary resuscitation; IV = intravenous; IO = intraosseous.

**Table 4. CPR outcomes for patients in internal medicine departments (N = 360).**

| Variable | CPR Team (N = 198) | Ward Team (N = 162) | P Value |
|---|---|---|---|
| ROSC N (%) | 96 (48.5) | 45 (27.8) | <0.001 |
| Survival 24h N (%) | 45 (22.7) | 25 (15.4) | 0.082 |
| Survival 30d N (%) | 17 (8.6) | 6 (3.7) | 0.060 |
| Survival at discharge N (%) | 15 (7.6) | 3 (1.9) | 0.013 |
| LOS, days (median, inter-quartile range) | 6 (2–13) | 6 (2–13) | 0.556 |
| CPC< = 2 N (%) | 8 (4.0) | 3 (1.9) | 0.230 |

This table describes the absolute numbers of patients with each characteristic collected from the electronic medical records. The number in brackets represents the percentage of patients out of n, n being the total number of patients in the group (CPR team vs. ward team). For continuous variables, a limit was set and the absolute number was counted accordingly. P-value was calculated for each variable when comparing the groups, and is shown here. CPR = cardiopulmonary resuscitation; ROSC = return of spontaneous circulation; LOS = length of stay; CPC = cerebral performance category.

compressions and early defibrillation [29, 30]. A recently published scoping review of the literature even suggests that low-dose high-frequency retraining every 1–6 months may be more beneficial than less frequent training, mostly explained by the rapid decay in CPR knowledge and skills after retraining sessions [31].

Interestingly, we found a significant difference between the groups in only two interventions during CPR–intubation and placing IV access–both significantly more frequent in the CPR team. This can explain the significant difference in crude ROSC and survival-to-discharge rates. However, the lack of significant difference in mortality in 24 hours and 30 days, in length of stay and in survival-to-discharge with propensity-score analysis implies that this difference in management is not necessarily associated with beneficial outcomes for the patients.

The importance of a rapid response team in hospitals was already introduced in previous studies [3, 32], but its definition, composition, and training differ between hospitals. Surprisingly, there are only a few recommendations published on how CPR teams should be constructed. A large study in the United States [33] and another one in Denmark [34] both demonstrate wide variability in team composition, including different team sizes, different qualifications of team members, rapid team turnover, and lack of senior supervision. These factors may explain the variability in survival results between different hospitals. In addition, Kyu Oh et al. [35] compared between CPRs conducted by resident teams, emergency medicine teams and rapid response teams in Seoul National University Bundang Hospital, and showed significantly higher ROSC rates and 10-day-survival in patients treated by rapid response teams, but no significant difference in 30-day survival and survival-to-discharge compared to CPRs conducted by resident teams. These findings support our findings, but do not take into account the teams' ACLS certification nor the variability between different departments and qualifications within the hospital. Our study suggests that frequent ACLS retraining (specifically every two to three years, with a few hours dedicated to full practice simulations), combined with the established communication routes of cohesive ward teams, can provide the same results as a CPR conducted by a team of experienced cardiologists and anesthesiologists.

Nallamothu et al. [36] conducted a qualitative study to characterize successful CPR teams at top-performing hospitals, and came up with four main themes: team design; team composition and roles; communication and leadership; and training and educational efforts. According to this study, hospitals with dedicated or designated teams (core groups without any other clinical responsibilities during a given shift, or with some responsibilities that can be dismissed when

**Table 5. Logistic regression for CPR team call (for propensity score analysis) in Internal medicine departments (N = 360).**

| Variable | | OR | 95% CI | P value |
|---|---|---|---|---|
| Age >75 | | 0.89 | 0.52–1.53 | 0.672 |
| Gender | Female | 0.79 | 0.48–1.29 | 0.346 |
| Ethnicity | Bedouin (with Jew as reference group) | 0.87 | 0.41–1.87 | 0.720 |
| Charlson comorbidity index ≥4 | | 1.14 | 0.42–3.09 | 0.795 |
| Dementia | | 0.50 | 0.25–0.99 | 0.048 |
| s/p CVA | | 0.53 | 0.29–0.95 | 0.033 |
| Malignancy | | 0.62 | 0.34–1.14 | 0.123 |
| Norton (admission) ≥15 | | 0.87 | 0.27–2.78 | 0.807 |
| CPR primary physician (with intern/ resident as reference group) | Attending | 0.34 | 0.17–0.70 | 0.003 |
| CPR hour (11pm-7am as reference group) | 7am-3pm | 0.25 | 0.13–0.51 | <0.001 |
| | 3pm-11pm | 0.75 | 0.41–1.36 | 0.335 |
| Norton (CPR day) ≥15 | | 2.01 | 0.62–6.57 | 0.248 |
| Mechanical ventilation prior to CPR | | 0.19 | 0.08–0.45 | <0.001 |
| Vasopressors prior to CPR | | 0.67 | 0.29–1.53 | 0.339 |
| CPR rhythm (with PEA/ asystole as reference group) | pulseless VT/VF/ other | 0.90 | 0.45–1.82 | 0.771 |

This table describes the odds ratio for each variable with respect to survival-to-discharge after an event of in-hospital cardiac arrest.

CVA = cerebrovascular accident; CPR = cardiopulmonary resuscitation; PEA = pulseless electrical activity;

VT = ventricular tachycardia; VF = ventricular fibrillation.

needed, respectively) perform better than hospitals with ad-hoc teams, some of which have no CPR teams at all [37]. However, according to Nallamothu et al., a major factor of their success is team members' acquaintanceship and frequent mock codes training. Although our study was not designed to examine specific communication, leadership, training, and medical education for CPR teams, the lack of significant difference shown when comparing designated CPR teams with internal ward teams may support the idea that previous familiarity of team members and a well-recognized leadership (e.g., attending physicians, head of department) play an important role in CPR outcomes.

It should be noted, that in order to address the subjective nature of deciding whether or not to call the dedicated CPR team, we looked into and compared different characteristics of the patients, both on admission and on the day of CPR. The lack of significant difference in co-morbidities and frailty scores, as well as in vital signs and lab results on admission, suggests similarity in baseline characteristics that usually affect disease prognosis. Lack of significant difference is shown also when comparing vital signs and lab results on the day of CPR,

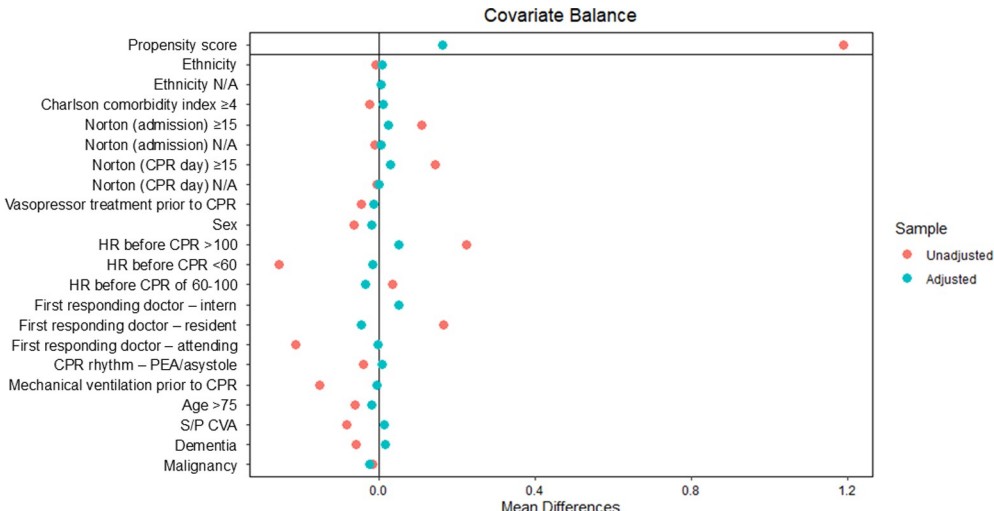

**Fig 2. Covariate balancing propensity score.** The figure illustrates the balance of different covariates after adjustment using propensity score weighting. N/A = non-applicable; CPR = cardiopulmonary resuscitation; HR = heart rate; PEA = pulseless electrical activity; S/P CVA = status-post cerebrovascular accident.

intensive care unit (ICU) admission prior to CPR, number of patients who underwent CPR during the first 24 hours after admission, and the average time in days from admission to CPR. This suggests similarity in severity of illness between the groups, strengthening our ability to compare outcomes. However, patients who were managed by ward teams had a significantly lower Norton score and a significantly higher prevalence of low blood pressure and fever, which can be confounders that partially influence the lower survival rate in this group. Nevertheless, adjusting to these factors did not change the outcome.

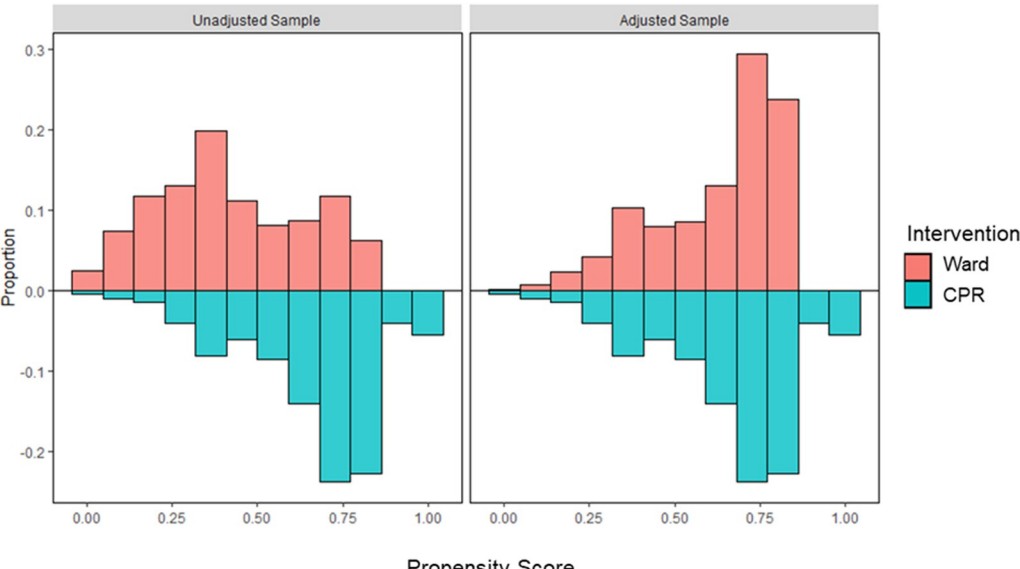

**Fig 3. Distributional balance for propensity score.** The figure illustrates the covariate distributions between the patients treated by the ward team (red) and the patients treated by the designated CPR team (blue) in both the adjusted sample (right) and the unadjusted sample (left).

In addition, we used propensity score weighting in order to minimize the possible selection bias when physicians initiating CPR decides whether or not to call the hospital's CPR team. One may argue that physicians are more reluctant to call the hospital's CPR team when they assume that the patient's probability of survival is low. Indeed, Table 5 shows that calling the hospital's CPR team is inversely correlated with a patient's history of dementia or cerebrovascular accident, when an attending physician starts the CPR and when the patient is already mechanically ventilated. Therefore, this selection bias may explain the significant difference in crude rates of survival-to-discharge after a CPR managed by the hospital's CPR team compared to ward team (Table 4). This difference becomes non-significant after using propensity score analysis. However, due to the fact that only three patients (1.9%) treated by ward teams survived to discharge, and that the RR was relatively high (1.97), this study may lack the power to achieve statistical significance.

It should also be noted, that due to cultural, religious and legal issues, a do-not-resuscitate (DNR) order is uncommon in Israel. Patients with high comorbidity index and therefore poor prognosis to survive IHCA may have a DNR order in Europe or North America, hence they would not undergo CPR. This is not the case in Israel, and even more so in its southern region, in which our hospital is located [38]. This issue may explain the relatively low overall survival rate of IHCA in our study compared to others [1–5]. Furthermore, almost every internal medicine ward in Israel contains an intermediate care unit for mechanically ventilated patients due to shortage in intensive care unit beds [39]. Therefore, all internal medicine residents are required to receive at least a one-month rotation and training in intensive care during their residency. In our study, all of the reported patients who were mechanically ventilated prior to CPR were hospitalized in these intermediate care units and none of them survived the CPR, regardless of the responding team.

Lastly, this study is not free of limitations. The observational and retrospective nature of this study makes it prone to documentation errors and missing data, though the SUMC electronic medical record database follows the standardized regulations and guidelines of the Israeli Ministry of Health. The study represents the experience of a single medical center, so generalization of the conclusions may be difficult. Finally, we do not have data on the quality of CPR performed, which can also affect the results.

## Conclusion

We found no difference in survival between IHCA treated by a dedicated hospital CPR team compared to a standard ward team, both trained with biennial ACLS training. This may suggest that frequent training and good team communication is of greater importance than team composition and expertise. Yet, it should be noted that crude survival-to-discharge was significantly higher in the CPR-team group. Further research is needed to finalize better recommendations regarding team composition and roles, and minimizing selection bias.

## Author Contributions

**Conceptualization:** Yasmeen Abu Fraiha, Ori Galante.

**Data curation:** Yasmeen Abu Fraiha, Alexander Fichman, Ori Galante.

**Formal analysis:** Tali Shafat, Ori Galante.

**Investigation:** Yasmeen Abu Fraiha, Ori Galante.

**Methodology:** Yasmeen Abu Fraiha, Tali Shafat, Dror Dolfin, Jacob Dreiher, Ori Galante.

**Project administration:** Yasmeen Abu Fraiha, Amit Frenkel, Yuval Konstantino, Said Abu Abed, Doron Schwartz, Alexander Fichman, Luba Kvich, Ori Galante.

**Supervision:** Shlomi Codish, Yuval Konstantino, Ori Galante.

**Validation:** Amit Frenkel, Dror Dolfin, Jacob Dreiher, Yuval Konstantino, Doron Schwartz, Ori Galante.

**Writing – original draft:** Yasmeen Abu Fraiha.

**Writing – review & editing:** Yasmeen Abu Fraiha, Shlomi Codish, Amit Frenkel, Dror Dolfin, Jacob Dreiher, Yuval Konstantino, Ori Galante.

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
