## [Decision Letter · Decision Letter 0]

18 Jun 2024

PONE-D-24-16710Outcomes of In-Hospital Cardiac Arrest Managed with and without a Specialized Code Team: A Retrospective Observational StudyPLOS ONE

Dear Dr. Abu Fraiha,

Thank you for submitting your manuscript to PLOS ONE. After careful consideration, we feel that it has merit but does not fully meet PLOS ONE’s publication criteria as it currently stands. Therefore, we invite you to submit a revised version of the manuscript that addresses the points raised during the review process.

We look forward to receiving your revised manuscript.

Kind regards,

Jignesh K. Patel

Academic Editor

PLOS ONE

Journal Requirements:

2. In the online submission form, you indicated that [Insert text from online submission form here]. 

Additional Editor Comments:

The authors paper examines cardiac arrest outcomes with or without the utilization of dedicated CPR teams.

Please address following questions:

1) was there any data showing discrepancies in management between the two groups?

2)It was acknowledged that the relative risk of survival to discharge was high and may not be adequately powered. Would the authors be able to reach the presented conclusion with such limitations? Is there any additional data or thought as to why the CPR team was activated in some cases and not others?

Reviewers' comments:

Reviewer's Responses to Questions

**Comments to the Author**

1. Is the manuscript technically sound, and do the data support the conclusions?

Reviewer #1: Yes

2. Has the statistical analysis been performed appropriately and rigorously? 

Reviewer #1: Yes

3. Have the authors made all data underlying the findings in their manuscript fully available?

Reviewer #1: Yes

4. Is the manuscript presented in an intelligible fashion and written in standard English?

Reviewer #1: Yes

5. Review Comments to the Author

Reviewer #1: I commend the authors on writing this paper as it has the potential to contribute to practice changes at many institutions.

This paper examines cardiac arrest outcomes with or without the utilization of dedicated CPR teams.

In regards to whether a designated CPR team was activated or not: the possible selection bias was addressed in the discussion. It was acknowledged that the relative risk of survival to discharge was high and may not be adequately powered. Would the authors be able to reach the presented conclusion with such limitations? Is there any additional data or thought as to why the CPR team was activated in some cases and not others?

On the other hand, all participants were ACLS trained - was there any data showing discrepancies in management between the two groups? It does state there is no data on CPR quality. If this data is available, it can further support the notion that ACLS retraining likely takes precedent, and therefore having dedicated specialists perform ACLS protocol likely provide no additional benefits.

This paper is well written, has clear objectives, and addresses significant previous published literature. Specifically the paper written by Nallamothu et al. which was published in Circulation was reviewed, and the authors provided an explanation on why such conclusions from that paper were reached.

This study is impactful by emphasizing that ACLS retraining and team familiarity/dynamics are likely of key importance of cardiac arrest management, rather than having dedicated CPR teams. This can support institutions to direct their efforts on ACLS retraining and team dynamics, rather than allocating significant resources to specifically design a dedicated CPR team.

6. PLOS authors have the option to publish the peer review history of their article (what does this mean?). If published, this will include your full peer review and any attached files.

Reviewer #1: **Yes: **Frank Yasnowski

---

## [Author Response · Author response to Decision Letter 0]

20 Jul 2024

Point-by-Point Responses to the Reviewers' Comments

We would like to thank the editor and the reviewers for their insightful comments. We respectfully submit our revised manuscript with the following changes, and with the attempt to fully address every comment made by the reviewers.

Journal Requirements:

We have gone thoroughly over the style requirements, and rearranged the manuscript accordingly.

2. In the online submission form, you indicated that [Insert text from online submission form here]. All PLOS journals now require all data underlying the findings described in their manuscript to be freely available to other researchers, either 1. In a public repository, 2. Within the manuscript itself, or 3. Uploaded as supplementary information.

The data that was used in this study is owned by Clalit Health Services, the organization that owns and operates Soroka University Medical Center, the hospital in which this study was conducted. Unfortunately, according to the organization’s data policy, we are not allowed to publicly publish any databases or specific information regarding patients, even if it is anonymized and without any identifying data. For that reason, the authors have no permission to upload the original database to any platform online.

We have thoroughly revised the reference list and made sure that it is complete and correct. No changes were made in the manuscript regarding references.

Additional Editor Comments:

The authors paper examines cardiac arrest outcomes with or without the utilization of dedicated CPR teams.

Please address following questions:

1) was there any data showing discrepancies in management between the two groups?

We thank the reviewer for this clever remark. We actually do have this data, and decided previously to take it out of the final paper due to lack of space. However, as it is proven important to the discussion, we have added the information regarding interventions during CPR to Table 3. We found a significant difference between the groups in only two interventions – intubation and placing IV access, both significantly more frequent in the CPR team. This can explain the significant difference in crude ROSC and survival-to-discharge rates. However, the lack of significant difference in mortality in 24 hours and 30 days, in length of stay and in survival-to-discharge with propensity-score analysis implies that this difference in management is not necessarily associated with beneficial outcomes for the patients. We have added this statement to the discussion in page 18, lines 355-360.

2) It was acknowledged that the relative risk of survival to discharge was high and may not be adequately powered. Would the authors be able to reach the presented conclusion with such limitations? Is there any additional data or thought as to why the CPR team was activated in some cases and not others?

We thank the reviewer for this excellent remark. We do acknowledge that there might be a selection bias when deciding to call the hospital’s CPR team. Due to the retrospective design of this study, it is impossible to know what made certain physicians activate the CPR team. We tried to address different possibilities. For example, one possible explanation may be that the primary team believed that the patient’s prognosis is poor to begin with, so there is no reason to activate the CPR team. Table 1 shows that there is no significant difference in baseline characteristics between the groups (except for Norton score), rebutting this possible selection bias. Another possible explanation may be that experienced attending physicians may tend to activate the CPR team less then residents and interns. In order to deal with this possibility, as well as with other factors that may impact the physician’s decision to activate the CPR, we’ve created a propensity score model to address this possible selection bias (explained in pages 19-20, lines 389-411). The propensity score model was designed according to the possibility to activate the CPR team per different factors, and this analysis did not result in a statistically significant difference in survival-to-discharge rate (Table 5, Fig 2 and 3).

It should be noted, that the sentence in the paper that addresses the possibility of lack of power to show statistically significant difference (page 20, lines 409-411) is for the propensity score analysis only, as it was powered enough to show a statistically significant difference in crude survival-to-discharge rates (Table 4). Therefore, we’ve tried to phrase our conclusion and discussion very carefully, stating that we did not find a significant difference in survival, although we acknowledge the importance of a hospital dedicated rapid response team and the possible explanation for this finding may be the similar ACLS training and retraining that all teams are going through. We hope that this explanation sheds more light on our discussion and conclusion in the paper.

Reviewers' comments:

1. Is the manuscript technically sound, and do the data support the conclusions?

Reviewer #1: Yes

We thank the reviewer for this kind comment.

2. Has the statistical analysis been performed appropriately and rigorously?

Reviewer #1: Yes

We thank the reviewer for this kind comment.

3. Have the authors made all data underlying the findings in their manuscript fully available?

Reviewer #1: Yes

We thank the reviewer for this kind comment. As mentioned earlier, the data that was used in this study is owned by Clalit Health Services, the organization that owns and operates Soroka University Medical Center, the hospital in which this study was conducted. Unfortunately, according to the organization’s data policy, we are not allowed to publicly publish any databases or specific information regarding patients, even if it is anonymized and without any identifying data. For that reason, the authors have no permission to upload the original database to any platform online.

4. Is the manuscript presented in an intelligible fashion and written in standard English?

Reviewer #1: Yes

We thank the reviewer for this kind comment.

5. Review Comments to the Author

Reviewer #1: I commend the authors on writing this paper as it has the potential to contribute to practice changes at many institutions.

This paper examines cardiac arrest outcomes with or without the utilization of dedicated CPR teams.

In regards to whether a designated CPR team was activated or not: the possible selection bias was addressed in the discussion. It was acknowledged that the relative risk of survival to discharge was high and may not be adequately powered. Would the authors be able to reach the presented conclusion with such limitations? Is there any additional data or thought as to why the CPR team was activated in some cases and not others?

We thank the reviewer for this excellent remark. We do acknowledge that there might be a selection bias when deciding to call the hospital’s CPR team. Due to the retrospective design of this study, it is impossible to know what made certain physicians activate the CPR team. We tried to address different possibilities. For example, one possible explanation may be that the primary team believed that the patient’s prognosis is poor to begin with, so there is no reason to activate the CPR team. Table 1 shows that there is no significant difference in baseline characteristics between the groups (except for Norton score), rebutting this possible selection bias. Another possible explanation may be that experienced attending physicians may tend to activate the CPR team less then residents and interns. In order to deal with this possibility, as well as with other factors that may impact the physician’s decision to activate the CPR, we’ve created a propensity score model to address this possible selection bias (explained in pages 19-20, lines 389-411). The propensity score model was designed according to the possibility to activate the CPR team per different factors, and this analysis did not result in a statistically significant difference in survival-to-discharge rate (Table 5, Fig 2 and 3).

It should be noted, that the sentence in the paper that addresses the possibility of lack of power to show statistically significant difference (page 20, lines 409-411) is for the propensity score analysis only, as it was powered enough to show a statistically significant difference in crude survival-to-discharge rates (Table 4). Therefore, we’ve tried to phrase our conclusion and discussion very carefully, stating that we did not find a significant difference in survival, although we acknowledge the importance of a hospital dedicated rapid response team and the possible explanation for this finding may be the similar ACLS training and retraining that all teams are going through. We hope that this explanation sheds more light on our discussion and conclusion in the paper.

On the other hand, all participants were ACLS trained - was there any data showing discrepancies in management between the two groups? It does state there is no data on CPR quality. If this data is available, it can further support the notion that ACLS retraining likely takes precedent, and therefore having dedicated specialists perform ACLS protocol likely provide no additional benefits.

We thank the reviewer for this clever remark. We actually do have this data, and decided previously to take it out of the final paper due to lack of space. However, as it is proven important to the discussion, we have added the information regarding interventions during CPR to Table 3. We found a significant difference between the groups in only two interventions – intubation and placing IV access, both significantly more frequent in the CPR team. This can explain the significant difference in crude ROSC and survival-to-discharge rates. However, the lack of significant difference in mortality in 24 hours and 30 days, in length of stay and in survival-to-discharge with propensity-score analysis implies that this difference in management is not necessarily associated with beneficial outcomes for the patients. We have added this statement to the discussion in page 18, lines 355-360.

This paper is well written, has clear objectives, and addresses significant previous published literature. Specifically the paper written by Nallamothu et al. which was published in Circulation was reviewed, and the authors provided an explanation on why such conclusions from that paper were reached.

This study is impactful by emphasizing that ACLS retraining and team familiarity/dynamics are likely of key importance of cardiac arrest management, rather than having dedicated CPR teams. This can support institutions to direct their efforts on ACLS retraining and team dynamics, rather than allocating significant resources to specifically design a dedicated CPR team.

We thank the reviewer for these thorough positive comments.

6. PLOS authors have the option to publish the peer review history of their article (what does this mean?). If published, this will include your full peer review and any attached files.

We appreciate this possibility, and would love to publish the peer-review history for this article together with the original paper.

---

## [Editor Report · Decision Letter 1]

12 Aug 2024

Outcomes of In-Hospital Cardiac Arrest Managed with and without a Specialized Code Team: A Retrospective Observational Study

PONE-D-24-16710R1

Dear Dr. Abu Fraiha,

We’re pleased to inform you that your manuscript has been judged scientifically suitable for publication and will be formally accepted for publication once it meets all outstanding technical requirements.

Kind regards,

Jignesh K. Patel

Academic Editor

PLOS ONE

Additional Editor Comments (optional):

Authors have made appropriate changes to the manuscript.
---

## [Editor Report · Acceptance letter]

11 Sep 2024

PONE-D-24-16710R1 

PLOS ONE

Dear Dr. Abu Fraiha, 

I'm pleased to inform you that your manuscript has been deemed suitable for publication in PLOS ONE. Congratulations! Your manuscript is now being handed over to our production team.

Kind regards, 

on behalf of

Dr. Jignesh K. Patel 

Academic Editor

PLOS ONE